# Prepregnancy BMI, gestational weight gain and offspring caries experience: Avon longitudinal study of parents and children

Aderonke A. Akinkugbe[1,2]*, Tegwyn H. Brickhouse[1,2], Dipankar Bandyopadhyay[3], Marcelle M. Nascimento[4], Gary D. Slade[5]

1 Department of Dental Public Health and Policy, School of Dentistry, Virginia Commonwealth University, Richmond, Virginia, United States of America, 2 Institute for Inclusion, Inquiry, and Innovation, Virginia Commonwealth University, Richmond, Virginia, United States of America, 3 Department of Biostatistics, School of Medicine, Virginia Commonwealth University, Richmond, Virginia, United States of America, 4 Department of Restorative Dental Sciences, University of Florida College of Dentistry, Gainesville, Florida, United States of America, 5 Pediatric Public Health Department, Adams School of Dentistry, University of North Carolina at Chapel Hill, Chapel Hill, North Carolina, United States of America

* aaakinkugbe@vcu.edu

**Data Availability Statement:** The ALSPAC website contains detail of all available data through a fully searchable data dictionary and variable search tool

## Abstract

Pre-existing maternal overweight/obesity and pregnancy weight gain are associated with adverse birth outcomes such as low birth weight and prematurity, which may increase the risk of developmental tooth defects and early childhood caries. We sought to investigate the association between prepregnancy BMI, gestational weight gain (GWG) and the risk of early childhood caries. Data from 1,429 mother-offspring participants of the 1991/1992 Avon Longitudinal Study of Parents and Children were analyzed. The exposures were prepregnancy BMI (under/normal weight vs. overweight/obese), and gestational weight gain (GWG) based on the Institute of Medicine's recommended levels. The main outcome measured was offspring caries experience determined by clinical oral examinations at three time points. Log binomial regression estimated risk ratios and 95% confidence intervals. Seventy six percent (76%) of the mothers were under/normal weight prepregnancy, 39% and 26% respectively gained less and more than the recommended weight for their prepregnancy BMI during pregnancy. Being overweight/obese prepregnancy was associated with unadjusted RR (95% CI) of offspring caries of 1.16 (0.90, 1.51) at 31-months, 1.20 (0.96, 1.49) at 43-months and 1.09 (0.91, 1.30) at 61-months. GWG less than recommended was associated with higher unadjusted offspring caries experience of 1.13 (0.86, 1.48), 1.17 (0.92, 1.48) and 1.04 (0.87, 1.25) at 31-months, 43-months and 61-months respectively. There was insufficient evidence to indicate an association between prepregnancy BMI and gestational weight gain on offspring caries experience risk.

## Introduction

The prevalence of overweight and obesity are on the rise in the U.S. and around the world [1]. The 2004–2005 pregnancy risk assessment and monitoring system indicate that 23% of

that can be accessed at: http://www.bristol.ac.uk/alspac/researchers/our-data/. Data are not publicly available but requests can be made from the ALSPAC Institutional Data Access / Ethics Committee (contact via alspac-exec@bristol.ac.uk) for researchers who meet the criteria for access to confidential data which can be found at http://www.bristol.ac.uk/alspac/researchers/.

**Funding:** This work was supported by the National Institutes of Health/National Institute of Dental and Craniofacial Research (Grant No.: R03DE028403 and L40DE028120) to A.A. https://www.nidcr.nih.gov/ The views expressed are solely the authors and does not represent the official views of the NIH/NIDCR. The funding source was not involved in the design, analysis, interpretation or decision to submit this article for publication.

**Competing interests:** The authors have declared that no competing interests exist.

pregnant U.S. women were overweight and 19% were obese [2]. More recent data from 2014 birth certificate records found that 26% and 25% of pregnant U.S. women were overweight and obese respectively [3], thus making pre-gravid overweight and obesity frequent high-risk obstetrics clinical conditions [4]. Pregnancy weight is reported to be associated with preterm birth and childhood obesity in the offspring, with prenatal care costs for overweight women about 5.4–16.2 times those of normal weight women [5].

The prenatal period is critical to the developmental origins of health and diseases [6]. Indeed, exposures occurring in the prenatal period are increasingly recognized as having significant impacts on later health outcomes [7]. Thus, the sensitivity of the developing fetus to maternal environment is not a new concept as demonstrated by the well-established impacts of smoking on adverse pregnancy outcomes [8–10] and pregnancy BMI on adverse offspring general health outcomes [11, 12].

Overweight/obesity alters the gut microbiome [13] and induces low grade inflammation [14]. Prenatal smoking promotes local oral inflammation [15], and likewise alters the oral microbiome [13] and induces low-grade systemic inflammation [16]. Thus, prenatal smoking and pregnancy adiposity likely share common biochemical and cellular mechanisms that promote an intrauterine environment conducive to adverse health outcomes (including dental caries) in the offspring.

Maternal overweight and obesity during pregnancy are associated with adverse birth outcomes and have been reported in at least one study to exceed smoking as a risk factor with the greatest risk of adverse pregnancy outcomes [17]. Moreover, maternal pregnancy adiposity [12] is associated with low-birth weight and prematurity, conditions that have been reported to increase the risk of developmental enamel defects [18, 19], that are more susceptible to cariogenic bacteria [20, 21] and consequently the occurrence of dental caries. Early childhood caries is the most prevalent chronic childhood disease and it negatively affects the oral health-related quality of life of children and their families at all socioeconomic levels but especially in low socioeconomic levels [22]. Furthermore, poor oral health early in childhood is a risk factor for continued poor oral health throughout the lifecourse [23].

A registry-based study previously reported associations between pregnancy BMI in the first trimester and subsequent caries experience in the teenage offspring [24]. Maternal health and lifestyle habits, such as diet, physical activity, weight, and cigarette smoking, can affect the child's oral health behaviors [25] and subsequent oral health status. Knowledge on associations between maternal health prepregnancy and in the prenatal period on dental caries occurrence in preschool age children are limited. Thus, this study aims to investigate the association between prepregnancy BMI, gestational weight gain and the risk of early childhood caries.

## Materials and methods

### Data source and study population

The Avon Longitudinal Study of Parents and Children (ALSPAC), is a prospective population-based birth cohort study originally aimed at studying environmental, and genetic factors affecting the health and development of children. ALSPAC recruited pregnant women residents of Avon, UK, with expected date of delivery between April 1991 and December 1992. Of 14,541 pregnancies enrolled, 13,761 mothers with singleton live births participated in the study [26–28]. A random 10% sample of children born in the last 6 months (June-December 1992) of the study were invited to participate in a sub-study called the "children in focus" (CIF) study. The CIF sample attended research clinics at approximate 6-month intervals during the first 5 years of life and at ages 31, 43 and 61 months, they underwent dental examination [26, 29] performed by dentists and health examiners. Trainings for the health examiners

took place in six tutorial sessions totaling 16 hours of training, accompanied by an hour-long session of mock examination and replication on 30 children. Reported kappa statistics for these health examiners was 0.63 [29].

The current study was restricted to the CIF sample because of available data on clinically determined dental caries status. Written informed consent was obtained at the time of data collection following the recommendations of the ALSPAC Ethics and Law Committee. The current study was approved by the Institutional Review Board at Virginia Commonwealth University as exempt (#:HM20011742) and reporting adhered to the Strengthening the Reporting of Observational studies in Epidemiology (STROBE) guidelines.

## Exposures

**Gestational weight gain.** Trained research midwives abstracted every measurement of weight entered into obstetric medical records and the corresponding gestational age and date. There was no between-midwife variation in mean values of abstracted data, and repeat data entry checks demonstrated error rates consistently lower than 1%. Absolute weight gain was calculated as the difference between the first and the last weight measurements in pregnancy [11]. For this investigation, the gestational weight gain (GWG) categories adopted of less than recommended, recommended and more than recommended, were based on the Institute of Medicine's (IOM) [30] recommended levels of GWG according to prepregnancy BMI categories. The recommended weight gain levels based on prepregnancy BMI are as follows: underweight—12.5 to 18kg; normal weight—11.5 to 16kg; overweight—7–11.5kg; obese—5-9kg (S1 Table in S1 File).

**Pre-pregnancy BMI.** Measures of prepregnancy weight in the parent ALSPAC cohort were predicted from abstracted pregnancy weight measurements using multilevel models as described by Fraser A et al., [11] while maternal height was self-reported. We calculated BMI according to the WHO specification and modeled it as continuous for descriptive purposes and categorized into underweight ($<18.5 kg/m^2$), normal ($18.5–24.9 kg/m^2$), overweight ($25–29.9 kg/m^2$) and obese ($\geq 30 kg/m^2$) in regression modeling.

## Outcome

**Offspring dental caries.** Using the World Health Organization criteria [29, 31] to assess the number of decayed, missing and filled teeth (dmft), dentists and health examiners conducted oral examinations at three time points: 31mo., 43 mo. and 5 years old. The outcome of caries experience for the current study was analyzed as dmft $\geq 1$ (yes vs. no) at 31 months, 43 months and 61 months old. Given this binary specification and to minimize outcome misclassification, we manually assigned missing dmft (yes vs. no) at each time point when clinically plausible. For instance, if dmft at a previous time point (say 31 mo.) classified a child as diseased (dmft $\geq 1$ i.e. 'yes'), we assigned the child as diseased at subsequent time points even if the child did not attend subsequent clinics or attended the clinic but was not examined. However, we did not assign subsequent case status to children lost to follow-up whose initial examination status indicated no disease since we could not determine if the child truly remained disease free or progressed to a diseased status. In instances where a later time point (say 61 mo.) indicated no disease (dmft = 0 i.e. 'no'), the corresponding prior time points (i.e. 31 mo. and 43 mo.) were manually assigned a non-disease status. In instances where a later time point indicated disease, previous time points were left as missing, since we could not determine when disease occurred. This manual dmft assignment described herein assumes that there were no errors in dental examinations. We multiply imputed missing dmfts that we could not manually assign a value to using chained equations [32] as described under the multiple imputation section.

## Covariates

**Maternal specific factors.** Were self-reported or abstracted from the medical records [27], and included maternal age at birth- modeled as continuous; education (≤O level, A-level, and college degree); maternal race (white vs. non-white); and breastfeeding duration (never, <6 months, ≥6months). Other variables include family structure (number of moves in the last 5 years); gestational age; mode of delivery (spontaneous vs. other) and prenatal smoking (never, any smoking during pregnancy).

**Child specific factors.** Were based on maternal report on mailed-in child specific questionnaires or determined during clinical evaluation of the CIF sample [26, 28]. These include child gender (male, female), history of childhood/perinatal illness (chicken pox, measles, or rubella), tooth brushing frequency (< 2 times daily vs. ≥2 times daily) and past year dental visit (yes vs. no) at 38 or 54 months. Other child specific factors included total sugar intake at 18, 43 and 61 months based on estimates calculated by the parent ALSPAC study from a series of 3-day food diaries for the CIF sample kept by their mothers. Based on our hypothesized causal diagram (S1 Fig in S1 File), these child-specific factors represent risk factors for caries experience in the offspring and to avoid over fitting our regression models, not all were included in our list of adjustment covariates. The ALSPAC website contains detail of all available data through a fully searchable data dictionary and variable search tool that can be accessed at: http://www.bristol.ac.uk/alspac/researchers/our-data/.

## Statistical analysis

Data analysis was restricted to singleton births (97%) who were alive at age 1 year. Data analysis began by summarizing the distribution of the exposures and baseline covariates for the CIF sample, reporting means and SDs for continuous variables and frequencies and relative frequencies for categorical variables. Using the Dagitty software (http://www.dagitty.net/) we identified a sufficient adjustment set of covariates (confounders) from a directed acyclic graph (DAG) [Akinkugbe et al., 2016] drawn apriori with knowledge from existing literature of causal relationships among variables presented on the DAG. Log-binomial regression estimated the 31-month, 43-month and 5- year risks of objectively determined offspring dental caries experience (i.e. dmft ≥1) and the corresponding risk ratios (RR) and 95% confidence intervals (C.I). The log-binomial model independently assessed the association between prepregnancy BMI groups and GWG categories on offspring caries experience at 31 months, 43 months and 61 months old. Analysis of prepregnancy BMI as the main exposure did not adjust for GWG because GWG lie on a causal pathway between prepregnancy BMI and offspring caries experience. Analysis of GWG as the main exposure adjusted for prepregnancy BMI as it represents a confounder of GWG and offspring caries experience.

**Multiple imputation.** Missing data were multiply imputed using chained equations (MICE) [White et al., 2011]. When there are several variables with missing data points, MICE is a practical approach to generate imputations based on a set of imputation models, one for each variable with missing values. Variables included in the imputation model were variables included in the final outcome models and predictors of missing data and loss to follow-up.

The following variables with missing observations were imputed: the number of decayed, missing and filled teeth at 31, 43 and 61 months; covariates with missing observations, as well as the exposures (prepregnancy weight, height and absolute weight gain). Eighteen percent of the IOM recommended gestational weight gain variable was missing. This variable is a composite of total weight gain during pregnancy and prepregnancy BMI. Since we independently imputed weight gain during pregnancy, we re-estimated the IOM gestational weight gain categories after multiple imputation. Refer to S2 Table in S1 File for the proportion of missing

covariates that we imputed. We imputed a total of 40 datasets using 500 between imputation iterations. Multiple imputation was done with PROC MI and MIANALYZE procedures in SAS that assumes data are missing at random (MAR). We summarized the results from each imputed dataset using Rubin's rule [Rubin, 1987] into an overall estimate that accounts for both within and between imputation variances and conducted all analyses in SAS v.9.4 (SAS Institute, Cary NC).

## Results

A total of 1,429 mother-child pairs were included in this study. Significant differences were noted in the distribution of certain maternal characteristics among the children in focus (CIF) mothers and the not in CIF mothers. Specifically, the CIF mothers were less likely to smoke during pregnancy (22% vs. 30%; p <0.0001), slightly older (29 vs. 28 years; p = 0.002), and more likely to have advanced degrees (40% vs. 35%; p = 0.001) than the not in CIF mothers. There were no significant differences with respect to race (p = 0.4), the number of moves in the past 5 years (p = 0.6) and absolute weight gain during pregnancy (p = 0.3) or in the proportion with chronic health conditions such as gestational diabetes (p = 0.6) and preeclampsia (p = 0.4). CIF mothers were less likely to be of normal prepregnancy BMI (71% vs. 75%) and more likely to be overweight (16% vs. 15%) and obese (7% vs. 5%) than the not in CIF mothers (p = 0.05). The CIF were more likely to be male (54% vs. 51%; p = 0.02) and white (96% vs. 95%; p = 0.01) than their counterparts not included in the CIF. However, there was no difference in gestational age at delivery between these groups (p = 0.2) Table 1. Three percent of the CIF had dmft ≥1 at 31 months, 16% at 43-months and 31% had dmft ≥1 at 61 months.

Among the CIF mothers, the mean (SD) prepregnancy BMI was 23 (4.1) kg/m$^2$, with 5% being underweight, 71% were of normal weight, 16% and 7% respectively were overweight and obese. The majority gave birth spontaneously (75%), and 34% had gestational weight gain within the IOM recommended level for their prepregnancy BMI category, 39% had less than the recommended GWG level and 26% more than the recommended level (Table 1).

Being overweight/obese prepregnancy was associated with greater unadjusted offspring caries experience risk at 31-months, 43-months and 61-months. Specifically, the caries experience risk for offspring of mothers who were overweight/obese prepregnancy was 21% at 31-months, 27% at 43-months and 36% at 61-months as compared to 18%, 23% and 33% for their underweight/normal weight counterparts at 31-months, 43-months and 61-months respectively. Similarly, caries experience risk was higher for mothers with less and more than the IOM recommended GWG levels as compared to those in the recommended weight gain category. For instance, the offspring caries experience risk was 18% at 31-months for those in the recommended GWG group as compared to 20% in the <recommended GWG group and 19% in the >recommended GWG group (Table 2).

While our unadjusted and adjusted findings indicated greater 31-month, 43-month and 61-month caries experience risk among offspring of mothers that were overweight/obese prepregnancy, and offspring of mothers who gained less or more than the recommended weight during pregnancy, none of the associations were statistically significant. Furthermore, the associations were strongest for caries experience at 43-months than either of the other two time points, with the adjusted estimates at 61-months being null. For instance, being overweight/obese prepregnancy was associated with unadjusted RR (95% CI) of offspring caries at 31-months of 1.16 (0.90, 1.51), 1.20 (0.96, 1.49) at 43-months and 1.09 (0.91, 1.30) at 61-months (Table 2). The corresponding adjusted estimates were attenuated towards the null and remained statistically non-significant. For instance, prepregnancy overweight/ obesity was

**Table 1. Distribution of selected covariates between children selected and those not selected for the Children-in-focus (CIF) sub-study: Avon longitudinal study of parents and children.**

| | CIF sample (n = 1,429) | Not in CIF sample (n = 13,449) | p-value |
|---|---|---|---|
| **Maternal characteristics** | | | |
| Prenatal Smoking | 305 (22) | 3,300 (30) | <0.0001 |
| Missing | 51 | 2,562 | |
| Age at delivery, mean (IQR) | 29 (26, 32) | 28 (25, 31) | 0.002 |
| Age at delivery (yrs.) | | | <0.0001 |
| 15–24 | 240 (17) | 3,098 (25) | |
| 25–35 | 1,072 (75) | 8,588 (69) | |
| 36–44 | 116 (8) | 851 (7) | |
| Missing | 1 | 912 | |
| Education | | | 0.001 |
| O-level or less | 819 (60) | 7,201 (65) | |
| A-level | 344 (25) | 2,443 (22) | |
| College degree | 199 (15) | 1,400 (13) | |
| Missing | 67 | 2,405 | |
| Race | | | 0.4 |
| White | 1,333 (98) | 10,662 (97) | |
| Non-white | 31 (2) | 292 (3) | |
| Missing | 64 | 2,495 | |
| Moves in the last 5 years | | | 0.6 |
| none | 341 (25) | 2,741 (24) | |
| 1–2 | 729 (53) | 6,064 (54) | |
| ≥3 | 297 (22) | 2,581 (23) | |
| missing | 62 | 2,063 | |
| Prepregnancy BMI mean (SD) | 23 (4.1) | 23 (3.8) | 0.003 |
| Underweight | 67 (5) | 514 (5) | 0.05 |
| Normal | 909 (71) | 7,672 (75) | |
| Overweight | 209 (16) | 1,524 (15) | |
| Obese | 87 (7) | 558 (5) | |
| Missing | 157 | 3,181 | |
| Absolute weight gain during pregnancy mean (SD) | 12.4 (4.8) | 12.5 (4.8) | 0.3 |
| Method of delivery | | | 0.4 |
| Spontaneous | 1067 (75) | 9,185 (76) | |
| Other | 356 (25) | 2,901 (24) | |
| missing | 6 | 1,363 | |
| Pre-eclampsia | | | 0.4 |
| Yes | 92 (7) | 858 (7) | |
| No | 1,332 (93) | 11,178 (93) | |
| missing | 5 | 1,413 | |
| Gestational diabetes | | | 0.6 |
| Yes | 10 (1) | 100 (1) | |
| No | 1,418 (99) | 12,199 (99) | |
| Missing | 1 | 1,150 | |
| Gestational weight gain | | | 0.7 |
| <recommended | 459 (39) | 3,529 (39) | |
| Recommended | 403 (34) | 3,050 (34) | |
| >Recommended | 259 (26) | 2,490 (27) | |

(*Continued*)

**Table 1.** (Continued)

| | CIF sample (n = 1,429) | Not in CIF sample (n = 13,449) | p-value |
|---|---|---|---|
| missing | 259 | 4,380 | |
| Gestational weight gain >40lbs | | | 0.5 |
| No | 1,203 (90) | 9,792 (89) | |
| Yes | 140 (10) | 1,219 (11) | |
| missing | 86 | 2,438 | |
| **Child Characteristics** | | | |
| Gender | | | 0.02 |
| Male | 772 (54) | 6,828 (51) | |
| Female | 657 (46) | 6,621 (49) | |
| Race | | | |
| White | 1,293 (96) | 10,169 (95) | 0.01 |
| Non-white | 47 (4) | 562 (5) | |
| Missing | 89 | 2,718 | |
| Gestational age at delivery mean (SD) | 39 (1.7) | 39 (1.9) | 0.2 |

CIF- Children in Focus; absolute weight gain-difference between last and first weight measurement

Estimates are N (%) unless otherwise noted

associated with the following adjusted offspring caries experience risk: 1.11 (0.85, 1.45) at 31-months, 1.12 (0.90, 1.39) at 43-months and 1.01 (0.85, 1.21) at 61-months (Table 3).

Children of mothers with GWG less than recommended had higher unadjusted caries experience at the three time points studied than children of mothers in the recommended GWG category. The RR (95% CI) for caries experience at 31-months was 1.13 (0.86, 1.48), 1.17 (0.92, 1.48) at 43-months and 1.04 (0.87, 1.25) at 61-months (Table 2). The corresponding estimates for children of mothers with GWG more than recommended were respectively 1.04 (0.78, 1.40), 1.14 (0.91, 1.50) and 1.13 (0.93, 1.36). The adjusted estimates were likewise attenuated towards the null and remained statistically non-significant (Table 3).

**Table 2. Risks and relative risks of the associations between pre-pregnancy BMI, gestational weight gain and offspring caries experience: Avon longitudinal study of parents and children (= 1,429).**

| | | 31 months Unadjusted | | | 43 months Unadjusted | | | 61 months Unadjusted | | |
|---|---|---|---|---|---|---|---|---|---|---|
| | Total N | N cases | Ris* | RR (95% CI)* | N cases | Risk* | RR (95% CI)*** | N cases | Risk* | RR (95% CI)*** |
| **Pre-pregnancy BMI** | | | | | | | | | | |
| Underweight/ ≤Normal | 1,091 | 199 | 0.182 | Ref. | 249 | 0.228 | Ref. | 359 | 0.329 | Ref. |
| Overweight/Obese | 338 | 72 | 0.213 | 1.16 (0.90, 1.51) | 92 | 0.27 | 1.20 (0.96, 1.49) | 121 | 0.358 | 1.09 (0.91, 1.30) |
| **GWG (IOM)** | | | | | | | | | | |
| <recommended | 494 | 100 | 0.203 | 1.13 (0.86, 1.48) | 125 | 0.253 | 1.17 (0.92, 1.48) | 165 | 0.333 | 1.04 (0.87, 1.25) |
| recommended | 545 | 98 | 0.179 | Ref. | 118 | 0.216 | Ref. | 174 | 0.320 | Ref. |
| >recommended | 390 | 73 | 0.187 | 1.04 (0.78, 1.40) | 98 | 0.252 | 1.14 (0.91, 1.50) | 141 | 0.361 | 1.13 (0.93, 1.36) |
| **GWG (>40lbs)** | | | | | | | | | | |
| No | 1,279 | 246 | 0.192 | Ref. | 309 | 0.241 | Ref. | 426 | 0.333 | Ref. |
| Yes | 150 | 25 | 0.167 | 0.90 (0.61, 1.33) | 32 | 0.216 | 0.89 (0.64, 1.25) | 55 | 0.363 | 1.09 (0.86, 1.38) |

*unadjusted risks

All estimates were averages from 40 rounds of multiple imputation combined using Rubin's rule and the variance a function of the within and between completed dataset variances.

**Table 3. Adjusted associations between pre-pregnancy BMI, gestational weight gain and offspring caries experience: Avon longitudinal study of parents and children (N = 1,429).**

| | 31 months | 43 months | 61 months |
|---|---|---|---|
| | Adjusted RR (95% CI)* | Adjusted RR (95% CI)*** | Adjusted RR (95% CI)*** |
| **Pre-pregnancy BMI** | | | |
| Underweight/normal | Ref. | Ref. | Ref. |
| Overweight/obese | 1.11 (0.85, 1.45) | 1.12 (0.90, 1.39) | 1.01 (0.85, 1.21) |
| **GWG (IOM)** | | | |
| <recommended | 1.10 (0.84, 1.42) | 1.13 (0.90, 1.42) | 1.02 (0.85, 1.22) |
| = recommended | Ref. | Ref. | Ref. |
| >recommended | 1.01 (0.76, 1.35) | 1.12 (0.87, 1.44) | 1.02 (0.92, 1.33) |
| **GWG (>40lbs)** | | | |
| No | Ref. | Ref. | Ref. |
| Yes | 0.90 (0.61, 1.33) | 0.94 (0.67, 1.32) | 1.13 (0.90, 1.41) |

Adjusted for maternal age, education, child gender, Tooth brushing frequency, method of delivery, hypertension, diabetes, race, prenatal smoking, and number of moves in the last 5 years

All estimates were averages from 40 rounds of multiple imputation combined using Rubin's rule and the variance a function of the within and between completed dataset variances

## Discussion

The current study found insufficient evidence to indicate an association between prepregnancy BMI and gestational weight gain on the risk of offspring caries experience. Our study focused on prepregnancy BMI and gestational weight gain as opposed to prenatal BMI as done by other studies that found an association. Indeed, Julihn et al., reported caries increment among teenagers of mothers who were overweight during pregnancy [24] and a higher caries experience risk in 3 and 7 year old Swedish children exposed to prenatal obesity [33]. Likewise Wigen et al., reported maternal overweight and obesity to be a risk indicator for caries experience among 5 year old Norwegian children [25]. A study of 6 year old, mono and dizygotic twins [34] likewise reported positive associations between maternal obesity and cavitated carious lesions and with concordance between monozygotic and dizygotic twins, indicating that shared and non-shared environmental factors predominate over genetic factors in determining variation in caries risk in children. On the contrary, Un Lam C et al., in a study of prenatal, perinatal and postnatal predictors of early childhood caries failed to identify maternal BMI during the prenatal period as a risk factor for ECC of 2-year-old children [35]. Of note many of the studies that reported significant associations did not adjust for caries risk factors such as sugar intake, tooth brushing frequency, method of delivery (that have the potential to affect the oral microbiome composition and subsequent caries risk), breast feeding duration except for Silva MJ et al., [34] and Un Lam C et al., [35].

The relationship between maternal BMI and offspring dental caries is complex because it is difficult to determine if caries risk is due to biological influences on the child or developing fetus, or a transfer of dietary habits, or perhaps confounding by social, environmental and other unknown factors.

Furthermore, the intrauterine environment of maternal obesity may lead to epigenetic changes that result in fetal programming that could result in future susceptibility to dental caries [36]. Pregnancy is an ideal time to focus on health promoting activities before the onset of disease given that most women have regular interactions with health care professionals [7]. High maternal prepregnancy BMI increases the risk of offspring obesity [11, 37], which is a

risk factor for caries experience in the offspring [35, 38, 39]. The weak association we found for prepregnancy BMI and offspring caries may thus, be a reflection of prepregnancy BMI being a more distal risk factor in comparison to obesity in the offspring.

Of note, our study assessed weight gain during pregnancy according to levels recommended by the IOM for different prepregnancy BMI groups. We found gestational weight gain less and more that recommended was associated with greater offspring caries experience risk, although not statistically significant. Presumably, an underweight woman may gain more than the recommended levels for her prepregnancy BMI group to put her into a normal BMI group or perhaps an overweight BMI group during pregnancy. Given that the BMI group a woman might fall into is likely to vary, this may also partly explain the inconclusive and sometimes null findings that we found that contradicts findings from previous studies that reported a positive association between prenatal BMI and offspring caries experience. Prepregnancy obesity and excessive gestational weight gain are risk factors for fetal macrosomia, cesarean delivery, hyperinsulinemia in infancy, and metabolic syndrome in childhood [13]. Maternal weight exceeding 200 pounds and gestational weight gain of more than 40 pounds have each been found to be associated with increased risk of autism and other developmental disabilities in the child [40]. Thus, we also conducted additional analysis of gestational weight gain in excess of 40 pounds on the risk of ECC and the results were similarly inconclusive as the main analysis suggests (Table 2).

While the predictors of ECC by Un Lam C et al., [35], found biological factors such as maternal age, and education more so than psychobehavioral factors, such as tooth brushing frequency as strong predictors of ECC, the current study adjusted for toothbrushing frequency and sugar intake as behavioral factors as well as maternal age, race, and education as adjustment covariates because they represent confounders of the association between prepregnancy BMI, gestational weight gain and offspring caries. Thus, their respective independent associations on the risk on ECC will require adjustments for different sets of covariates as potential confounders [41] and that was not the purpose of this study. Given that our study was not predictive in nature, we focused on reporting the findings of our main exposures, prepregnancy BMI, gestational weight gain associations on the risk of ECC.

## Limitations and strengths

Oral examination data was available only on a fraction of the children and the methodology used (i.e., dmft index) measured cavitated lesions and therefore likely to underestimate dental caries that are still in the incipient phase (white spot lesions) of development. Thus, if we had data on incipient caries lesions, the prevalence of ECC might have been higher and the reported association stronger. The non-diverse study population may limit statistical inference to other population subgroups but not necessarily the scientific inference of our findings [42]. Our sample size of 1,429 might not have been sufficiently large enough for us to detect meaningful effects especially given that prepregnancy BMI and gestational weight gain are more distal factors on the causal pathway to ECC and thus will have small effects that our small sample prevented us from detecting.

Study strengths include the longitudinal nature and the ability to minimize temporal ambiguity. Contemporaneous determination of weight during pregnancy allowed for the estimation of gestational weight gain. Furthermore, the availability of objectively assessed oral examination data at three time points represents another of this study's strengths. Lastly, we had access to several covariates on the mother and child that allowed us control for relevant confounding factors as well as risk factors for ECC in an attempt to minimize bias.

## Supporting information

**S1 File.**
(DOCX)

## Acknowledgments

We are extremely grateful to all the families who took part in this study, the midwives for their help in recruiting them, and the whole ALSPAC team, which includes interviewers, computer and laboratory technicians, clerical workers, research scientists, volunteers, managers, receptionists and nurses. The UK Medical Research Council, the Wellcome Trust (Grant ref: 217065/Z/19/Z) and the University of Bristol provide core support for ALSPAC. This publication is the work of the authors, and the authors will serve as guarantors for the contents of this paper.

## Author Contributions

**Conceptualization:** Aderonke A. Akinkugbe.

**Data curation:** Aderonke A. Akinkugbe.

**Formal analysis:** Aderonke A. Akinkugbe, Dipankar Bandyopadhyay.

**Funding acquisition:** Aderonke A. Akinkugbe, Tegwyn H. Brickhouse, Dipankar Bandyopadhyay.

**Investigation:** Aderonke A. Akinkugbe.

**Methodology:** Aderonke A. Akinkugbe, Tegwyn H. Brickhouse, Marcelle M. Nascimento, Gary D. Slade.

**Project administration:** Aderonke A. Akinkugbe.

**Supervision:** Marcelle M. Nascimento, Gary D. Slade.

**Writing – original draft:** Aderonke A. Akinkugbe.

**Writing – review & editing:** Aderonke A. Akinkugbe, Tegwyn H. Brickhouse, Dipankar Bandyopadhyay, Marcelle M. Nascimento, Gary D. Slade.

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
