## [Decision Letter · Decision Letter 0]

13 Dec 2021

PONE-D-21-29637Prepregnancy BMI, gestational weight gain and offspring caries experience: Avon Longitudinal Study of Parents and Children.PLOS ONE

Dear Dr. Akinkugbe,

Thank you for submitting your manuscript to PLOS ONE. After careful consideration, we feel that it has merit but does not fully meet PLOS ONE’s publication criteria as it currently stands. Therefore, we invite you to submit a revised version of the manuscript that addresses the points raised during the review process.

We look forward to receiving your revised manuscript.

Kind regards,

Angela Lupattelli, PhD

Academic Editor

PLOS ONE

Journal Requirements:

a) Did participants provide their written or verbal informed consent to participate in this study?

“The UK Medical Research Council, the Wellcome Trust (Grant ref: 217065/Z/19/Z) and the University of Bristol provide core support for ALSPAC.”

“This work was supported by the National Institutes of Health/National Institute of Dental and Craniofacial Research (Grant No.: R03DE028403 and L40DE028120) to A.A.  https://www.nidcr.nih.gov/

The views expressed are solely the authors and does not represent the official views of the NIH/NIDCR.  The funding source was not involved in the design, analysis, interpretation or decision to submit this article for publication.”

5. We note you have included a table to which you do not refer in the text of your manuscript. Please ensure that you refer to Table 3 in your text; if accepted, production will need this reference to link the reader to the Table.

Reviewers' comments:

Reviewer's Responses to Questions

**Comments to the Author**

1. Is the manuscript technically sound, and do the data support the conclusions?

Reviewer #1: Partly

Reviewer #2: Yes

2. Has the statistical analysis been performed appropriately and rigorously? 

Reviewer #1: Yes

Reviewer #2: Yes

3. Have the authors made all data underlying the findings in their manuscript fully available?

Reviewer #1: Yes

Reviewer #2: Yes

4. Is the manuscript presented in an intelligible fashion and written in standard English?

Reviewer #1: Yes

Reviewer #2: Yes

5. Review Comments to the Author

Reviewer #1: The authors have to be commented for looking into the ALSPAC data from a different perspective. However, I do have some concerns in regards to the correlations made without taking into consideration important co-founding factors. Below are my comments for improvement of the manuscript if it is selected for publication.

Introduction:

1. Generally in the manuscript many times the authors combine information in the same sentence on different topics. Editing of the manuscript, using shorter sentences and more references. For example, the last sentence of the first paragraph "Pregnancy weight....normal weight women" presents the effect of pregnancy weight on birth and childhood obesity and combines this information with treatment cost which is irrelevant. I think that many of the sentences need further development to conclude to the need for this study.

2. Many times the authors make speculations ex. maternal adiposity leads to prematurity that is associated with developmental defects which are more susceptible to cariogenic bacteria, meaning higher caries. The literature data does not suggest any of this. The authors should be more careful in their wording and present this as a speculation that will be investigated through this study.

3. The main findings from the literature that can state the argument for the aim of the study in the fourth paragraph are not developed.

4. The last paragraph should be omitted or should be placed higher in the introduction to help the argument for the need of this study.

Materials & Method (is not in a title in the manuscript)

1. I am concerned about the time that these data were collected (1991/1992). There have been many changes in the diet and caries risk of children, as well as to the level of obesity of pregnant women ever since and I think that these data is outdated.

2. Many times in the manuscript the age of collection data is presented in months and some times in months and years. There must be some consistency for the reader.

3. Dental caries is recorded as yes or no. White spot lesions are considered important for caries risk in preschool children. Thus, dental caries prevalence using yes or no is not sufficient for this age group.

4. Maternal caries is a very important factor in caries prevalence of preschool children and it was not recorded. In general, caries is a multifactorial disease if all these factors (maternal caries, dietary habits, toothbrushing, breastfeeding habits) are not put in the model I am not sure the results can be accurate. The authors report that toothbrushing, diet and breastfeeding were recorded but these data are not described in the results.

Results

I don't see why the data from the non-CIF should be presented in the manuscript since the aim is to correlate caries and nonCIF pairs did not have any caries data.

Discussion

1. Many of the factors were not discussed in the manuscript, as well as the many limitations of the study. The discussion is very limited and insufficient to support the findings of the study.

2. One of the main findings were the fact that less than recommended GWG was associated with higher caries. How do the authors explain this finding?

Reviewer #2: This study is one that analyzed the association between mother’s prepregnancy BMI and gestational weight gain, and children’s dental caries. Overall, this study is well organized. The viewpoint of authors is of interest. However, there are some points to improve.

Results

Data of caries status of subjects (dmf and rate of having caries) is not seen in this study.

Discussion

The influence of difference in mother’s characteristics between CIF sample and non-CIF sample should be discussed. For example, lower smoking and higher education might lower dental caries overall.

The date available in this study is old (expected date of delivery between 1991 and 1992). Authors must explain that this old data is still applicable to current state.

6. PLOS authors have the option to publish the peer review history of their article (what does this mean?). If published, this will include your full peer review and any attached files.

Reviewer #1: No

Reviewer #2: No

---

## [Author Response · Author response to Decision Letter 0]

22 Jan 2022

The authors thank the reviewers for the time and effort spent reviewing our manuscript: “PONE-D-21-29637, Prepregnancy BMI, gestational weight gain and offspring caries experience: Avon Longitudinal Study of Parents and Children”. Our response to reviewers’ comments appears below and in track changes in the actual manuscript draft. We have also reformatted our manuscript to adhere to the journal formatting requirements, updated the cover letter with funding information and data access policy.

Reviewers' comments:

Reviewer's Responses to Questions

Comments to the Author

1. Is the manuscript technically sound, and do the data support the conclusions?

Reviewer #1: Partly

Reviewer #2: Yes

2. Has the statistical analysis been performed appropriately and rigorously? 

Reviewer #1: Yes

Reviewer #2: Yes

3. Have the authors made all data underlying the findings in their manuscript fully available?

Reviewer #1: Yes

Reviewer #2: Yes

4. Is the manuscript presented in an intelligible fashion and written in standard English?

Reviewer #1: Yes

Reviewer #2: Yes

5. Review Comments to the Author

Reviewer #1: The authors have to be commented for looking into the ALSPAC data from a different perspective. However, I do have some concerns in regards to the correlations made without taking into consideration important co-founding factors. Below are my comments for improvement of the manuscript if it is selected for publication.

Introduction:

1. Generally in the manuscript many times the authors combine information in the same sentence on different topics. Editing of the manuscript, using shorter sentences and more references. For example, the last sentence of the first paragraph "Pregnancy weight....normal weight women" presents the effect of pregnancy weight on birth and childhood obesity and combines this information with treatment cost which is irrelevant. I think that many of the sentences need further development to conclude to the need for this study.

Response: Thank you for your comment. The first paragraph the reviewer referred described the prevalence of overweight and obesity in pregnancy, the consequence of this on the offspring including the impact in terms of costs on the health care system. All of these point to or highlight the significance of studying this condition in pregnancy. 

2. Many times the authors make speculations ex. maternal adiposity leads to prematurity that is associated with developmental defects which are more susceptible to cariogenic bacteria, meaning higher caries. The literature data does not suggest any of this. The authors should be more careful in their wording and present this as a speculation that will be investigated through this study.

Response: Thank you for your comments. We made no speculations at all. The sentence the reviewer referred to is properly cited for the interested reader.

3. The main findings from the literature that can state the argument for the aim of the study in the fourth paragraph are not developed.

Response: Thank you for your comment. There is not a lot of literature out there to the authors knowledge that has investigated this topic. Based on the limited literature we are aware of and based on findings from a previous study on teenagers, we are conducting this study. The significance of the pregnancy weight and offspring dental caries are described in earlier paragraphs of the introduction section.

4. The last paragraph should be omitted or should be placed higher in the introduction to help the argument for the need of this study.

Response: We have added this paragraph to an earlier section of the introduction

Materials & Method (is not in a title in the manuscript)

1. I am concerned about the time that these data were collected (1991/1992). There have been many changes in the diet and caries risk of children, as well as to the level of obesity of pregnant women ever since and I think that these data is outdated.

Response: We agree that the data is dated. Nevertheless, the prevalence of dental caries as well as overweight and obesity in this dataset are comparable to contemporary US benchmarks. Please note that the purpose of this study is to determine to the extent possible and within the limitations of this data if overweight and obesity in pregnancy affects dental caries occurrence in the offspring. The is an etiological question that has nothing to do with the prevalence of these conditions.

2. Many times in the manuscript the age of collection data is presented in months and some times in months and years. There must be some consistency for the reader.

Response: Age at dental caries assessment has been updated throughout the manuscript to age in months

3. Dental caries is recorded as yes or no. White spot lesions are considered important for caries risk in preschool children. Thus, dental caries prevalence using yes or no is not sufficient for this age group.

We agree and we stated this in the limitations section. As a matter of fact, it is likely that dental caries is underestimated since the dmft index only measures cavitated lesions. The ALSPAC study has only measured dental cavitated lesions and not white spot lesion/incipient caries.

4. Maternal caries is a very important factor in caries prevalence of preschool children and it was not recorded. In general, caries is a multifactorial disease if all these factors (maternal caries, dietary habits, toothbrushing, breastfeeding habits) are not put in the model I am not sure the results can be accurate. The authors report that toothbrushing, diet and breastfeeding were recorded but these data are not described in the results.

Response: We agree that these are important variables, which we described in Table 1 and noted on our causal diagram S1 Fig. Based on our causal diagram in S1 Fig and the analysis of this causal diagram for potential confounders to adjust for, we adjusted for a sufficient covariate subset to block all confounding pathways and leave open all causal pathways. To avoid committing “Table 2 fallacy” we cannot independently report the findings of the variables the reviewers pointed out on dental caries since they were not the primary predictors. Please refer to Weistrech et al 2010 for more information on this fallacy

Results

I don't see why the data from the non-CIF should be presented in the manuscript since the aim is to correlate caries and nonCIF pairs did not have any caries data.

Response: This is to determine if the CIF and non-CIF mothers and children differ in any of their baseline covariates. i.e., to see if there are systematic differences between these groups. The data of the non-CIF were not utilized beyond what was described in Table 1.

Discussion

1. Many of the factors were not discussed in the manuscript, as well as the many limitations of the study. The discussion is very limited and insufficient to support the findings of the study.

Response: We discussed the findings of our analysis of whether pre-pregnancy BMI, and pregnancy weight are associated with offspring caries experience. Again, there is not a lot of literature out there to the authors knowledge to compare and contrast our findings with and we avoided making speculations beyond what an epidemiological study can tell us.

2. One of the main findings were the fact that less than recommended GWG was associated with higher caries. How do the authors explain this finding?

Response: Yes, the magnitude of the effect was small and not statistically significant

Reviewer #2: This study is one that analyzed the association between mother’s prepregnancy BMI and gestational weight gain, and children’s dental caries. Overall, this study is well organized. The viewpoint of authors is of interest. However, there are some points to improve.

Results

Data of caries status of subjects (dmf and rate of having caries) is not seen in this study.

Response: We reported this information in the results section as : Three percent of the CIF had dmft ≥1 at 31 months, 16% at 43-months and 31% had dmft ≥1 at 61 months.

Discussion

The influence of difference in mother’s characteristics between CIF sample and non-CIF sample should be discussed. For example, lower smoking and higher education might lower dental caries overall.

The date available in this study is old (expected date of delivery between 1991 and 1992). Authors must explain that this old data is still applicable to current state.

Response: While the prevalence of prenatal smoking (22%) in the ALSPAC cohort [Macdonald-Wallis et al., 2011] was higher than contemporary U.S. benchmarks of 9-14% [Kurti et al., 2017, Berlin and Oncken, 2018], 25% of 5 year-olds in the ALSPAC study experienced caries [Kay et al., 2010], as have similarly aged contemporary U.S. children [Dye et al., 2015]. 

Oral examination data was available only on a fraction of the children and the methodology used (i.e. dmft index) measured cavitated lesions. This is in contrast to contemporary methods like the international caries detection and assessment system (ICDAS) that records dental caries on a continuum that includes incipient lesions. In spite of this, the underlying biologic mechanism for the proposed association is undated and applicable even today.

6. PLOS authors have the option to publish the peer review history of their article (what does this mean?). If published, this will include your full peer review and any attached files.

Do you want your identity to be public for this peer review? For information about this choice, including consent withdrawal, please see our Privacy Policy.

Reviewer #1: No

Reviewer #2: No

---

## [Decision Letter · Decision Letter 1]

18 Feb 2022

PONE-D-21-29637R1Prepregnancy BMI, gestational weight gain and offspring caries experience: Avon Longitudinal Study of Parents and Children.PLOS ONE

Dear Dr. Akinkugbe,

Thank you for submitting your manuscript to PLOS ONE. After careful consideration, we feel that it has merit but does not fully meet PLOS ONE’s publication criteria as it currently stands. Therefore, we invite you to submit a revised version of the manuscript that addresses the points raised during the review process.

We look forward to receiving your revised manuscript.

Kind regards,

Angela Lupattelli, PhD

Academic Editor

PLOS ONE

Additional Editor Comments (if provided):

Dear authors,

multiple comments received in the first review round have not been addressed, eg amendment of the Discussion section and expansion on the Limitations of the study (Reviewer 1, comment 10). Please respond to these comments and make appropriate changes to the Discussion section, as recommended.

Reviewers' comments:

Reviewer's Responses to Questions

**Comments to the Author**

1. If the authors have adequately addressed your comments raised in a previous round of review and you feel that this manuscript is now acceptable for publication, you may indicate that here to bypass the “Comments to the Author” section, enter your conflict of interest statement in the “Confidential to Editor” section, and submit your "Accept" recommendation.

Reviewer #1: (No Response)

2. Is the manuscript technically sound, and do the data support the conclusions?

Reviewer #1: No

3. Has the statistical analysis been performed appropriately and rigorously? 

Reviewer #1: Yes

4. Have the authors made all data underlying the findings in their manuscript fully available?

Reviewer #1: Yes

5. Is the manuscript presented in an intelligible fashion and written in standard English?

Reviewer #1: Yes

6. Review Comments to the Author

Reviewer #1: Some corrections were made, however the authors did not response to the major limitations in the methodology and for this reason I do not think it is suitable for publication.

7. PLOS authors have the option to publish the peer review history of their article (what does this mean?). If published, this will include your full peer review and any attached files.

Reviewer #1: No

---

## [Author Response · Author response to Decision Letter 1]

15 Mar 2022

The authors thank the reviewers for the second round of review of our manuscript: “PONE-D-21-29637R1, Prepregnancy BMI, gestational weight gain and offspring caries experience: Avon Longitudinal Study of Parents and Children”. Our response to reviewers’ comments appears below and in track changes in the actual manuscript draft. Please note that the latest iteration of track changes is only for the reviewers remaining concerns and does not include track changes from the previous submission which has been disabled.

Comments to the Author

Additional Editor Comments (if provided):

Dear authors,

multiple comments received in the first review round have not been addressed, eg amendment of the Discussion section and expansion on the Limitations of the study (Reviewer 1, comment 10). Please respond to these comments and make appropriate changes to the Discussion section, as recommended.

Authors’ response: Pasted below are the specific comments related to the discussion section from the previous review and our response to those comments

Discussion

1. Many of the factors were not discussed in the manuscript, as well as the many limitations of the study. The discussion is very limited and insufficient to support the findings of the study.

Response: We discussed the findings of our analysis of whether pre-pregnancy BMI, and pregnancy weight are associated with offspring caries experience. Again, there is not a lot of literature out there to the authors knowledge to compare and contrast our findings with and we avoided making speculations beyond what an epidemiological study can tell us.

Updated response: Please note that the association between adjustment covariates and the outcome of ECC was not the primary purpose of this study. Because we have not accounted for the confounding structure between those adjustment covariates and ECC, discussing their individual association from a model of pre-pregnancy BMI, and pregnancy weight and ECC is fallacious (please refer to the paper by Westreich and Greenland 2013 for detail of this fallacy)

2. One of the main findings were the fact that less than recommended GWG was associated with higher caries. How do the authors explain this finding?

Response: Yes, the magnitude of the effect was small and not statistically significant

Updated response: The following text has been added to the discussion 

“The weak association we found for prepregnancy BMI and offspring caries may thus, be a reflection of prepregnancy BMI being a more distal risk factor in comparison to obesity in the offspring.”

“Of note, our study assessed weight gain during pregnancy according to levels recommended by the IOM for different prepregnancy BMI groups. We found gestational weight gain less and more that recommended was associated with greater offspring caries experience risk, although not statistically significant.”

Reviewers' comments:

Reviewer's Responses to Questions

Comments to the Author

1. If the authors have adequately addressed your comments raised in a previous round of review and you feel that this manuscript is now acceptable for publication, you may indicate that here to bypass the “Comments to the Author” section, enter your conflict of interest statement in the “Confidential to Editor” section, and submit your "Accept" recommendation.

Reviewer #1: (No Response)

2. Is the manuscript technically sound, and do the data support the conclusions?

Reviewer #1: No

3. Has the statistical analysis been performed appropriately and rigorously?

Reviewer #1: Yes

4. Have the authors made all data underlying the findings in their manuscript fully available?

Reviewer #1: Yes

5. Is the manuscript presented in an intelligible fashion and written in standard English?

Reviewer #1: Yes

6. Review Comments to the Author

Reviewer #1: Some corrections were made, however the authors did not response to the major limitations in the methodology and for this reason I do not think it is suitable for publication.

Response: The authors do not understand the specific concern that the reviewer has with our manuscript or the limitations with the methodology that we have not addressed. Nevertheless, we have expanded the discussion section and the limitation section in the revised manuscript.

We also want to point that the association between adjustment covariates and the outcome of ECC was not the primary purpose of this study. Because we have not accounted for the confounding structure between those adjustment covariates and ECC, discussing their individual association from a model of pre-pregnancy BMI, and pregnancy weight and ECC is fallacious (please refer to the paper by Westreich and Greenland 2013 for detail of this fallacy)

Furthermore, the following text has been added to the discussion 

“The weak association we found for prepregnancy BMI and offspring caries may thus, be a reflection of prepregnancy BMI being a more distal risk factor in comparison to obesity in the offspring.”

“Of note, our study assessed weight gain during pregnancy according to levels recommended by the IOM for different prepregnancy BMI groups. We found gestational weight gain less and more that recommended was associated with greater offspring caries experience risk, although not statistically significant.”

7. PLOS authors have the option to publish the peer review history of their article (what does this mean?). If published, this will include your full peer review and any attached files.

Do you want your identity to be public for this peer review? For information about this choice, including consent withdrawal, please see our Privacy Policy.

Reviewer #1: No

---

## [Editor Report · Decision Letter 2]

17 Mar 2022

Prepregnancy BMI, gestational weight gain and offspring caries experience: Avon Longitudinal Study of Parents and Children.

PONE-D-21-29637R2

Dear Dr. Akinkugbe,

We’re pleased to inform you that your manuscript has been judged scientifically suitable for publication and will be formally accepted for publication once it meets all outstanding technical requirements.

Kind regards,

Angela Lupattelli, PhD

Academic Editor

PLOS ONE

---

## [Editor Report · Acceptance letter]

23 Mar 2022

PONE-D-21-29637R2 

Prepregnancy BMI, Gestational Weight Gain And Offspring Caries Experience: Avon Longitudinal Study Of Parents And Children. 

Dear Dr. Akinkugbe:

I'm pleased to inform you that your manuscript has been deemed suitable for publication in PLOS ONE. Congratulations! Your manuscript is now with our production department. 

Kind regards, 

on behalf of

Dr. Angela Lupattelli 

Academic Editor

PLOS ONE